# Targeting Cellular Components of the Tumor Microenvironment in Solid Malignancies

**DOI:** 10.3390/cancers14174278

**Published:** 2022-09-01

**Authors:** Carmen Belli, Gabriele Antonarelli, Matteo Repetto, Luca Boscolo Bielo, Edoardo Crimini, Giuseppe Curigliano

**Affiliations:** 1Division of New Drugs and Early Drug Development for Innovative Therapies, European Institute of Oncology, IRCCS, Via Ripamonti 435, 20141 Milan, Italy; 2Department of Oncology and Haematology (DIPO), University of Milan, 20141 Milan, Italy

**Keywords:** cancer, immunity, microenvironment, immunotherapy

## Abstract

**Simple Summary:**

The tumor microenvironment comprises numerous different cellular players that engage transformed cancer cells and may exert pro- vs anti-tumor functions. Their crosstalk, functionality, as well as cell recruitment to cancer lesions, is chiefly dictated by cytokines/chemokines-receptor pairs and these axes can represent therapeutic vulnerabilities—as for immune checkpoint blockers. In this review, we recapitulate the main drivers of cellular TME dynamics, interactions, and functionality, mainly focusing on T lymphocytes, macrophages and cancer associated fibroblasts, also providing an outlook on state-of-the-art TME-targeting agents.

**Abstract:**

Cancers are composed of transformed cells, characterized by aberrant growth and invasiveness, in close relationship with non-transformed healthy cells and stromal tissue. The latter two comprise the so-called tumor microenvironment (TME), which plays a key role in tumorigenesis, cancer progression, metastatic seeding, and therapy resistance. In these regards, cancer-TME interactions are complex and dynamic, with malignant cells actively imposing an immune-suppressive and tumor-promoting state on surrounding, non-transformed, cells. Immune cells (both lymphoid and myeloid) can be recruited from the circulation and/or bone marrow by means of chemotactic signals, and their functionality is hijacked upon arrival at tumor sites. Molecular characterization of tumor-TME interactions led to the introduction of novel anti-cancer therapies targeting specific components of the TME, such as immune checkpoint blockers (ICB) (i.e., anti-programmed death 1, anti-PD1; anti-Cytotoxic T-Lymphocyte Antigen 4, anti-CTLA4). However, ICB resistance often develops and, despite the introduction of newer technologies able to study the TME at the single-cell level, a detailed understanding of all tumor-TME connections is still largely lacking. In this work, we highlight the main cellular and extracellular components of the TME, discuss their dynamics and functionality, and provide an outlook on the most relevant clinical data obtained with novel TME-targeting agents, with a focus on T lymphocytes, macrophages, and cancer-associated fibroblasts.

## 1. Introduction

The tumor microenvironment (TME) is a complex and dynamic system comprising cellular and non-cellular components with a well-recognized role in the regulation of cancer progression, response to therapies, and development of acquired and adaptive resistance mechanisms. Cellular components of TME include cells such as T and B lymphocytes, tumor-associated macrophages (TAM), dendritic cells (DC), natural killer (NK) cells, neutrophils, myeloid-derived suppressor cells (MDSC), stromal cells such as cancer-associated fibroblast (CAF), pericytes, and mesenchymal stromal cells. Moreover, non-cellular components include the extracellular matrix (ECM), secreted molecules such as growth factors, cytokines, chemokines, extracellular vesicles, blood, and lymphatic cells. All these components play a critical role in the activation of signals that allow cancer cells to grow, invade the surrounding tissues, and metastasize to distant organs. Tumor cells metastasizing to distant tissues may meet a favorable microenvironment (pre-metastatic niche), which is positively conditioned by circulating factors released by the primary tumor [1,2]. Given the critical role of the TME in cancer development and progression other than in the regulation of the efficacy of anticancer therapies, many strategies targeting TME have been significantly expanded in the last years. 

The topic of this review is to provide a comprehensive analysis of different cellular TME components in solid tumors, their dynamics, interactions, as well as of the different therapeutic strategies in clinical development, with a particular focus on T lymphocytes, macrophages, and cancer-associated fibroblasts. 

## 2. Immune System and Cancer Interplay

The immune system consists of a range of components including different organs and immune cells. These cells include macrophages, DC, mast cells, NK, naïve, and memory B-lymphocytes, T-helper (Th) cells mainly classified into Th1, Th2, and Th17 cells, regulatory T cells (Treg), and CD8+ cytotoxic T lymphocytes (CTLs). Immune cells can be found in the tumor core, in the invasive margin, or in the adjacent lymphoid structure. The activation of the immune system against cancers is a complex process requiring the initiation of different steps starting from recognizing tumor cells’ antigens, to effector T cell activation through two signals [3]. Signal 1 is delivered through the T cell receptor (TCR), after engagement by a peptide-MHC complex displayed on APCs or cancer cells. Signal 2, a co-stimulatory signal, requires multiple ligand-receptor interactions that transmit both positive and negative signals to T cells. 

CD8+ T cells are the most prominent anti-tumor effector cells. These cells, after interaction with APC, differentiate in CTLs, which directly kill cancer cells by releasing perforin and granzyme B [3]. In contrast, CD4+ Th1 cells regulate immune response against the tumor by producing pro-inflammatory cytokines namely interferon gamma (IFN-γ), tumor necrosis factor α (TNF-α), and interleukin 2 (IL-2), promoting not only the activation of CTLs but also the anti-tumor activity of macrophages, with enhanced antigen presentation [3]. 

In the TME, both inflammation and tumor cells’ antigen presentation can drive T cells towards an “exhausted” state that progressively loses effector function and proliferative capacity [4]. A major hallmark of T cell exhaustion is enhanced engagement of inhibitory signal by overexpression of multiple co-inhibitor checkpoint molecules such as programmed cell death receptor-1 (PD-1), cytotoxic T lymphocyte-associated protein 4 (CTLA-4), often co-expressed with other inhibitory receptors such as T-cell immunoglobulin and ITIM domain (TIGIT), or T-cell immunoglobulin-3 (TIM-3). These molecules interacting with their respective ligands, present on APC contribute to intratumoral T cell dysfunction [3]. Consequently, inhibition of these checkpoint proteins can rev up the immune response against tumors. In T cell exhaustion, a prominent role is also carried out by stromal cells as tumor-associated dendritic cells, Treg, TAM, and MDSC [1]. Treg cells are a subset of CD4+ T cells suppressing the function of T cell expression by up-regulation of co-inhibitory checkpoint molecules, by the generation of adenosine and production of cytokines such as IL-10 and Transforming Growth Factor beta (TGF-β) suppressing both maturation and expression of co-stimulatory factors on APC [5,6]. Additional soluble factors, growth factors, and chemokines can contribute to the immune escape mechanism by inhibiting maturation of DC, migration of T cells in the tumor bed other than CD8+ cells apoptosis giving evidence of the complexity and dynamism of TME and the need to understand these interactions to implement combined therapeutic strategies for cancer management [1].

## 3. Targeting Immune System in Cancer

The treatment of cancer has undoubtedly been changed with the introduction of monoclonal antibodies targeting the immune system. Since 2011, when ipilimumab was first approved for the treatment of metastatic melanoma, many other immunotherapeutic agents have been approved alone or in combination with chemotherapy (Figure 1), and currently, there are more than 2000 active clinical trials testing T cell modulators in cancer [7]. Apart from CTLA-4 and the PD-1/PD-L1, new checkpoint inhibitors have been discovered which can be targeted by specific monoclonal antibodies (mAb). These drugs are under investigation in clinical trials as monotherapy or in combination with other agents and will also be the topic of our discussion. 

### 3.1. T Lymphocytes

**LAG-3**: is a transmembrane glycoprotein receptor found on the surface of effector T-cells, T-reg cells, NK cells, activated B-cells, and DCs [8]. The major ligand is MHC-II, but other ligands (i.e., α-synuclein, galectin, LSECtin, and FGL1) have also been described. The interaction between LAG-3 and its ligand inhibits TCR with reduced cytokine production and T cell proliferation [9]. LAG-3 is a co-inhibitory receptor overexpressed, together with other co-inhibitory molecules such as PD-1, on exhausted T cells [10]. Despite similar functions, LAG3 and PD-1 activate different pathways downregulating the immune system and preclinical studies showed that inhibition of both signals decreases tumor growth and enhances antitumor activity [10]. 

Several LAG-3 inhibitors have been developed and are under evaluation as monotherapy or in combination with anti-PD1/antiPD-L1 therapies, including monoclonal antibody anti-LAG3, antagonistic bispecific antibodies targeting LAG-3 and PD-1 or targeting both LAG-3 and CTLA-4. Positive results are coming out from clinical trials testing the combination of anti-LAG3 with anti-PD1/antiPD-L1 therapies confirming data of preclinical studies showing a synergistic antitumoral action given by dual blockade of these immune checkpoint inhibitors [10]. 

Relatlimab: is the only anti-LAG3 mAb that got FDA approval in combination with nivolumab in untreated advanced melanoma patients according to the result of the RELATIVITY-047 trial [11,12] (Table 1). This phase II/III trial showed that the combination of relatlimab plus nivolumab, administered until disease progression or unacceptable toxicity, demonstrated an improvement in median progression-free survival (mPFS), the primary endpoint of the study, compared to monotherapy with nivolumab alone (mPFS 10.22 vs. 4.63, HR 0.75 95% CI 0.62–0.92, *p* = 0.006). Patients with characteristics associated with a worse prognosis, such as visceral metastases, high tumor burden, elevated levels of serum LDH, or mucosal/acral melanoma, had better outcomes with relatlimab–nivolumab than with nivolumab alone. Relatlimab–nivolumab combination was beneficial over nivolumab alone regardless of the BRAF status. Treatment-related adverse events (TRAEs) were more common in the combination arm compared to the monotherapy group (83.7 vs. 72.4%), with skin toxicity, thyroid dysfunction, arthralgia, and diarrhea as the most frequent TRAEs. Although cross-trial comparison should be undertaken with caution, the efficacy of relatlimab + nivolumab combination was seemingly better than nivolumab + ipilimumab [13]. Patient treated with relatlimab and nivolumab had lower rates of grade (G) 3/4 TRAEs (21.1% vs. 59%) and lower discontinuation rate (15.2% vs. 39%) compared to nivolumab + ipilimumab in CheckMate 067 trial [11,12,13]. The results of these trials supported the benefit of dual checkpoint inhibition in advanced melanoma and added another combination to the therapeutic armamentarium of this disease.

Relatlimab is also under investigation in phase I and II clinical trials in combination with other immune checkpoint inhibitors in different solid tumors (Table 1). 

Favezelimab: is an anti-LAG-3 inhibitor under investigation in combination with the PD-L1 inhibitor pembrolizumab in colorectal cancer (CRC) patients [14]. This trial enrolled patients with microsatellite stable (MSS) metastatic CRC progressing on ≥2 lines of therapies, who received favezelimab alone or pembrolizumab + favezelimab until disease progression or unacceptable toxicity. The preliminary results of this study showed promising activity of the combination pembrolizumab + favezelimab according to the combined positive score (CPS) of PD-L1. In the combination arm, the group with CPS score ≥ 1, in fact, had an ORR of 11% vs. 2.9% in the group with a CPS <1. Additionally, the tolerability profile was manageable in both arms with only 5.6% in the combination arm vs. 0% in the monotherapy group discontinuing the treatment due to TRAEs.

Tebotelimab (MGD013): belongs to bispecific moAb targeting both LAG-3 and PD-1 and it was tested in the phase I clinical trial alone or in combination with margetuximab (an anti-HER2 mAb with higher binding activity for activating Fcγ receptor CD16A and decreased affinity for inhibitory CD32B) in case of HER2 expression in advanced or metastatic solid tumors or hematologic malignancies [15]. Preliminary results of this trial are available and the ORR in 29 patients evaluable for response in dose-escalation part of the study was 10% and 7%, respectively, in 41 evaluable patients of the dose expansion cohort. Among the 6 patients with HER2 overexpression treated with margetuximab, ORR was 83% (5/6 patients). The most common AEs were G1/G2 fatigue with G3 AEs occurring in 23.2% of patients, consisting of rash, pancreatitis, and colitis.

**TIGIT**: this coinhibitory receptor is expressed on Treg, NK, and activated CD8+ and CD4+ T cells. It binds to two ligands, CD155 and CD112, which are both expressed on tumor cells and APC, and it displays multiple actions such as T cell inhibition by decreasing IL-12 production, NK activity inhibition, and Treg immunosuppressive function enhancement. Currently, several anti-TIGIT mAbs are under evaluation in clinical trials as monotherapy or in combination with anti-PD1/PD-L1 monoclonal antibodies (Table 1). 

Tiragolumab: is the anti-TIGIT mAb for which there are available data of activity coming out from the phase II clinical trial CTYSCAPE. In this study, patients with advanced non-small cell lung cancer (NSCLC) with EGFR/ALK wild type (WT) and PDL1 expression ≥ 1%, according to tumor proportion (TPS) score, received the combination of tiragolumab plus atezolizumab vs. atezolizumab alone. The mPFS, the primary endpoint of the study, was quadrupled in patients with tiragolumab therapy having a high PDL1 TPS score ≥ 50% compared to atezolizumab arm (16.6 vs. 4.1 months HR = 0.29 95% CI: 0.15–0.53), while no significant difference was reported in the group with low PD-L1 (4.0 vs. 3.6-month, HR= 1.7 95% CI 0.67–1.71). Additionally, a clinically relevant improvement in overall survival (OS) was reported in patients in the combination arm carrying high PD-L1 TPS score treated with the combination therapy compared with the monotherapy group (mOS not reached in combination arm vs. 12.8 months; HR = 0.23 95% CI: 0.10–0.53). The toxicity profile in the combination arm was manageable, with similar G2/G3 TRAEs in both arms (19.4% in the combination arm vs. 16.2% in the monotherapy group). Infusion reaction, skin rash, and fatigue were more common in patients treated with dual immune checkpoint blockade [16]. A caveat of this study was the small number of patients enrolled, thus the ongoing phase III trial SKYSCRAPER-01 (NCT04294810) is specifically designed to assess the efficacy of this combination as frontline therapy for NSCLC in a larger patient population with high PD-L1 score.

Vibostolimab: this anti-TIGIT mAb has demonstrated clinical activity in combination with pembrolizumab in a phase I trial conducted in different solid tumors [17]. This study consisted of two parts: part A enrolled patients affected with advanced solid tumors and part B enrolled NSCLC patients, naïve or refractory to immunotherapy. In part A, ORR was 0% in the monotherapy group vs. 7% in the combination group; in immune refractory NSCLC, no difference in ORR was reported (3% in both groups), while in NSCLC naïve group assigned only to the combination arm ORR was 26% in combination group regardless of PD-L1. The treatment was well tolerated in both arms of part A with G3/G4 TRAEs occurring in 9% of the monotherapy arm vs. 17% of the combination arm and consisted of skin rash and fatigue. In part B, the safety profile was similar with no G3/G4 AEs in the monotherapy arm vs. 15% in the combination group with infusion reaction, lymphopenia, hypophysitis, hypothyroidism, and pneumonia as the most common ones. The results of this study are limited by the modest sample size of patients enrolled but support further investigation.

**TIM-3**: is a member of a family of coinhibitory receptors that can be expressed both on tumor cells and multiple types of immune cells [18,19]. Different ligands that interact with TIM-3 have been identified, mainly galectin-9, phosphatidylserine, HMGB1, and CEACAM1 [19]. The interaction of TIM-3 on the effector T cell with its ligands inhibits the immune response by inducing T cell apoptosis. In addition, TIM-3 is also up-regulated on Treg cells, inhibiting T cell function and contributing to T cell exhaustion [20]. Overexpression of TIM-3 is predictive of poor OS for patients with NSCLC, gastric, colon, cervical, and clear renal cell carcinoma [21]. Several clinical trials evaluating anti-TIM-3 mAbs in cancer patients are currently underway and most of them are including a combination arm with PD1 inhibitors. 

Sabatolimab (MBG453): this anti-TIM-3 antibody is being tested in combination with spartalizumab (PDR001), a PD-1 inhibitor, in a phase I/II study in patients affected by solid tumors (NCT02608268) and compared to sabatolimab alone. No responses were reported in the monotherapy arm while 6% of 86 patients in the combination arm experienced a partial response lasting between 12 and 27 months. Responders included 2 patients with colorectal cancer and 1 each with NSCLC, malignant perianal melanoma, and small cell lung cancer. The most common TRAE was fatigue, reported in 9% of those who received monotherapy and in 15% receiving the combination [22].

**A2A adenosine receptor (A2AAR):** is a family of A G protein-coupled receptors, including four subtypes of adenosine receptors (A1, A2A, A2B, and A3), which are activated by extracellular adenosine [23]. The A1 and A2A receptors are widely distributed in both the central nervous system (CNS) and the periphery, while the density of A2B, and A3 in the brain is very low. Adenosine concentration increases rapidly in response to pathophysiological conditions such as hypoxia, ischemia, inflammation, or tissue injury [24], and the interaction with its receptor contributes to inhibiting the immune response in the TME by blocking effector T cell function and Treg activation [25]. Adenosine also promotes angiogenesis by inducing the secretion of the vascular endothelial growth factor (VEGF) [26]. Preclinical studies have shown that blocking A2AAR could reactivate the immune system and, in particular, the activity of CD8+ lymphocytes against the tumors [27]. The combination of A2AAR inhibiotrs with other checkpoint inhibitors led to enhanced antitumor activity [28].

A2AAR inhibitors are under evaluation in early-phase clinical trials as monotherapy or in combination with other checkpoint inhibitors, but the results of these studies are available only for a few drugs.

Ciforadenant (CPI-444): this selective A2AR was evaluated in a phase 1 clinical trial in a pretreated patient affected by renal cancer; this has been one of the first trials demonstrating that the inhibition of the adenosine pathway could be considered a therapeutic strategy in cancer immunotherapy [29]. Patients enrolled in this study received ciforadenant combined with atezolizumab vs. monotherapy. Median PFS was 4.1 months for ciforadenant monotherapy vs. 5.8 months for the combination group, with an estimated OS >95% at 16 months for the combo and >65% for the ciforadenant arm. Ciforadenant treatment was well tolerated, with G3/4 adverse events being infrequent for both arms of treatment (<3%). The results of this study were encouraging and supported the role of the adenosine receptor in cancer immunotherapy, but further research is needed to validate the effective role of this agent. 

BF-509 (or NIR-178): is a potent and selective A2AAR antagonist tested in a phase I/II study alone and in combination with anti-PD-1 mAb PDR001 in 25 patients with NSCLC (NCT02403193). The results of this study showed a favorable toxicity profile, with the most common G3 dose-limiting toxicities being nausea and liver function test elevation (occurring in 1 patient, each) and pneumonitis in the combination arm (2 patients), in immunotherapy-exposed and -naive patients regardless of the PD-L1 status. Responses were observed in 2 immunotherapy naïve patients, with stable disease in 21 patients, including 13 immunotherapy exposed patients [30]. 

### 3.2. Myeloid Cells

Myeloid cells are comprised of different innate immune cells, mainly granulocytes (neutrophils, basophils, eosinophils, mast cells), monocytes (Mo), macrophages (Mф), and dendritic cells (DCs). In the bone marrow (BM), a common myeloid progenitor (CMP), deriving from a hematopoietic stem and progenitor cell (HSPC), may differentiate either into a common DC progenitor or into a granulocyte-monocyte progenitor [31]. In turn, conventional (cDC) or plasmacytoid DC (pDC) originate from the former, whereas monocytes/macrophages and granulocytes originate from a granulocyte-monocyte progenitor. Myeloid cells represent a key cellular component of the TME, however, given their vast functional and developmental heterogeneity, they retain diverse prognostic/predictive values across different malignancies, with recognized functions related to tumor angiogenesis, antigen presentation, and therapy resistance [32]. Moreover, such heterogeneity also resulted in the lack of reliable and uniform markers to identify different myeloid subpopulations, resulting in the cumbersome identification of putative druggable targets, ultimately translating into largely negative therapeutic attempts so far. 

Macrophage-targeted therapies mainly consist either in their re-education, recruitment, depletion, or gene-engineering. In line with the scope of this review, we will discuss the clinical experience obtained with the modulation of myeloid cells’ recruitment in solid malignancies [33]. 

Myeloid-cells tumor homing is largely regulated by multiple ligand-receptor interactions, mainly the colony-stimulating factor-1 (CSF1R), the C-X-C Motif Chemokine Receptor 2 (CXCR2), the C-C chemokine receptor type 2 (CCR2) and type 5 (CCR5). In preclinical studies, targeting myeloid cell recruitment led to a significant reduction of tumor angiogenesis as well as to an increase in effector T cell functions, paving the way for clinical translation studies. 

**CSF1R axis inhibitors.** The CSF1R, also known as macrophage colony-stimulating factor receptor (M-CSFR), is expressed on myeloid cells, particularly by Mф. Pharmacologic targeting of this axis mainly relies on anti-CSF1 or anti-CSF1R agents as well as on CSF1R tyrosine kinase inhibitors (TKIs) and has shown anti-tumor activity in several pre-clinical models [33,34]. 

Pexidartinib, an orally bioavailable CSF1R TKI, has been approved in august 2019 by the food and drug administration (FDA) for the treatment of tenosynovial giant cell tumor (TGCT) in patients with severe morbidity or functional limitations not amenable to improvement by surgery. This approval was based on the results of the phase III ENLIVEN trial (NCT02371369), which showed an improved overall response rate (ORR) in patients receiving Pexidartinib compared to placebo of 39% vs. 0%, respectively (95% CI 27–53; *p* < 0.0001), with mixed/cholestatic hepatotoxicity representing a common treatment-related adverse event. In the MEDIPLEX trial, the combination of pexidartinib with durvalumab in 19 patients with advanced PDAC/CRC was generally well tolerated and associated with SD in two patients with MSI-H CRC [35]. Another member of this class is vimseltinib, an oral, switch control, TKI that inhibits the switch control region regulating the kinase conformational activation, and it is currently tested in the MOTION phase 3 clinical trial (NCT05059262) versus placebo in symptomatic patients affected by TGCT ineligible for surgery. Preclinical activity of vimseltinib alone, or in combination with anti-PD1 mAb, was shown, together with Mф depletion and suppression of CSF1R kinase activity. Moreover, in the phase I/II trial (NCT03069469), vimseltinib showed a tolerable safety profile, and biological as well as clinical activity in 13/29 patients [36]. 

Emactuzumab, instead, is an anti-CSF1R humanized murine mAb that blocks its dimerization, which showed a favorable toxicity profile but poor clinical activity in a phase I clinical study enrolling patients with advanced solid tumors (NCT01494688) [37,38]. Among patients with diffuse type TGCT, instead, emactuzumab therapy resulted in improved and durable clinical responses, as well as in quality-of-life assessment [38]. LY3022855, a human immunoglobulin G subclass 1 (IgG1) anti-CSF1R mAb, has been tested alone (NCT01346358) or in combination with tremelimumab (NCT02718911) in patients with advanced solid tumors, demonstrating limited clinical activity despite a good safety profile [39,40]. 

AMG820, a fully human anti-CSF1R IgG2 mAb, did not reach an MTD while demonstrating pharmacodynamic effects despite limited clinical activity when tested alone (NCT01444404) or in combination with pembrolizumab (NCT02713529) in patients with advanced solid tumors [41,42]. 

Cabiralizumab, a humanized IgG4 anti-CSF1R mAb, has also been tested in combination with nivolumab and radiotherapy in patients with locally-advanced pancreatic cancer (NCT03599362), in combination with anti-CD40 sotigalimab in ICB-refractory melanoma, non-small cell lung cancer, or kidney cancer patients, resulting in a PR in 1/26 patients and SD in 8/26 patients (NCT03502330), as well as in TGCT (NCT02471716) [43,44]. Pre- and on-treatment biopsies across multiple tumor types in patients being treated with cabiralizumab and nivolumab showed an increase of CD8 lymphocytes mirrored by a reduction of CD163+ CSF1R+ Mф (NCT02526017) [45]. Cabiralizumab is also currently being tested in the neoadjuvant setting with nivolumab in triple-negative breast cancer (NCT04331067), head and neck cancer (NCT04848116), and advanced hepatocellular carcinoma (NCT04050462). 

Lacnotuzumab (MCS110), an anti-CSF1 mAb, has been tested with carboplatin and gemcitabine in TNBC in a randomized phase II clinical trial. Interestingly, despite signs of CSF1 engagement, lacnotuzumab treatment resulted in similar clinical activity and poorer tolerability compared to chemotherapy alone [46]. In a phase Ib/II study (NCT02807844), lacnotuzumab + spartalizumab resulted in a disease control rate assessed by immune-related response criteria, of 27% (13/48) (irPR or irSD), with good tolerability in patients with advanced solid tumors [47]. 

Overall, anti-CSF1R therapies resulted in limited activity despite favorable pharmacodynamic profiles. Preclinical studies suggested various mechanisms of adaptive resistance upon CSF1R inhibition. These include the influx of granulocytic myeloid cells [48,49] or compensatory regulatory T cells [50] as well as the chemotactic effects of other cytokines present in the TME. In turn, CSF1R targeting has also been demonstrated to overcome resistance to PARP inhibitors in a BRCA1-associated TNBC model [51]. These data highlight the need for investigating suitable combinatorial treatment strategies with anti-CSF1R agents in order to reprogram the TME and support prolonged anti-tumor responses. 

**CCR2-CCL2 axis inhibitors.** The CCR2 receptor, found mainly on monocytes, dendritic and endothelial cells, binds to several ligands including CCL-2, -7, -8, and -13 and acts as a potent driver of monocyte mobilization [33,52]. 

Carlumab (CNTO888), a human IgG1κ mAb, has been tested in patients with metastatic castration-resistant prostate cancer (mCRPC), without evidencing objective, biochemical responses, or sustained CCL2 suppression (NCT00992186) [53]. Similar results were obtained in other solid tumors with carlumab monotherapy (NCT00537368) or in combination with chemotherapy (NCT01204996), leading to discontinuation of clinical testing. Similarly, another anti-CCR2 mAb, Plonalizumab, was tested in patients with melanoma in combination with ICB and no data have been published (NCT02723006).

PF-04136309, an oral CCR2 inhibitor, was evaluated in combination with FOLFIRINOX (5FU, leucovorin, irinotecan, oxaliplatin, NCT01413022) in patients with borderline resectable/locally advanced pancreatic cancer led to objective responses in 49% (16/33) patients and to a disease control rate of 97% (32/33). Grade 3 or more treatment-related toxicities developed in 10% of patients, with neutropenia being the most common [54]. Conversely, the combination of PF-04136309 with nab-paclitaxel/gemcitabine in untreated patients with metastatic pancreatic cancer (NCT02732938) showed signs of synergistic pulmonary toxicity without objective response improvements to chemotherapy alone, thus halting clinical development [55]. 

**CCR5–CCL5 axis inhibitors.** CCR5 is expressed on different cell types apart from Mф and DCs, including T lymphocytes and cancer cells (i.e., prostate and breast cancer), and mediates numerous functions such as immune suppression and angiogenesis [56]. 

Leronlimab (PRO 140), a humanized IgG4κ mAb, has been mostly tested in TNBC. In fact, in a preclinical model of TNBC, leronlimab use resulted in reduced cancer cell migration, invasion, and metastatic spreading [57,58]. Interestingly, a pooled analysis from the three clinical trials utilizing leronlimab in TNBC patients showed a drop in circulating tumor-associated cells in 21/28 patients, and signs of clinical activity, especially in patients receiving higher doses of leronlimab (525–700 mg), with mPFS of 6.1 months (95% CI 2.3–7.5) [59]. 

Maraviroc, an anti-CCR5 oral agent largely utilized in HIV patients [60]. In colorectal cancer (CRC), CCR5 blockade has been shown to induce Mф repolarization via STAT3 induction, leading to an increase of IFN-α2 and IFN-γ in the TME. A pilot trial (NCT01736813) investigating maraviroc in patients with refractory, metastatic, CRC, demonstrated a favorable safety profile and, among the five patients receiving maraviroc with chemotherapy, a disease control rate of 80% (4/5) [61].

BMS-813160, a potent and selective oral CCR2/5 dual antagonist, is currently being tested in combination with nivolumab in patients with HCC or NSCLC (NCT04123379) as well as in combination with nivolumab +/− GVAX immune-stimulatory vaccine in patients with PDAC [62].

**CXCR4 inhibitors**. The CXCR4-CXCL12 axis is involved in multiple cancer processes, including signaling pathways related to cellular proliferation (PI3K/Akt and Ras/Raf/MAPK) [63,64], immune-suppression [65,66,67], metastatic spreading [68,69] and angiogenesis [70]. Concerning the latter, it has been shown that CXCR4-CXCL12 axis inhibition curbs vasculogenesis in pre-clinical models through an AKT-mediated signaling pathway leading to VEGF induction. In addition, in mice, CXCR4 acts as a chemoattractant for circulating neutrophils to the liver, aiding in the formation of pre-metastatic niches [71]. Interestingly, the CXCR4-CXCL12 axis has been shown to interact with PD-1 on tumor cells, which is upregulated upon CXCR4 inhibition, promoting tumor cell invasiveness [72]. Combinatorial treatment strategies targeting both CXCR4 and PD-1 are currently being studied in several pre-clinical models [72,73,74,75] and clinical trials (NCT03168139, NCT04058145, NCT02826486, NCT03628859, NCT04177810, and NCT04177810).

Motixafortide (BL-8040), a small synthetic peptide targeting CXCR4, has been tested in patients with refractory, metastatic, pancreatic ductal adenocarcinoma (COMBAT, NCT02826486) in combination with pembrolizumab and chemotherapy. Of note, among the 22 patients enrolled in phase II, the ORR was 32% and DCR of 77%, with evidence of TME modulation, such as increased CD8+ effector T lymphocytes, decreased circulating Treg cells, and MDSCs [76]. Motixafortide is currently being tested in hematologic tumors (NCT02763384) and in solid tumors (NCT04543071, NCT03281369, NCT03193190).

Balixafortide (POL6326), a synthetic cyclic peptide, is being tested with nab-paclitaxel/eribulin in patients with HER2-negative, metastatic, refractory, breast cancer (NCT04826016, POLTER), after promising clinical activity demonstrated in a phase I trial (NCT01837095), with an ORR of 30% (95%CI, 18−44%) and good tolerability [77,78].

LY2510924, a peptide CXCR4 antagonist, showed a favorable toxicity profile in a phase I trial (NCT02737072), although no signs of efficacy were detected in the combination with sunitinib in patients with metastatic renal cell carcinoma (NCT01391130) [79], nor the combination of LY2510924 with chemotherapy improved objective responses in patients with extensive stage small cell lung cancer (NCT01439568) [80].

### 3.3. Cancer-Associated Fibroblasts and Extracellular Matrix

**Cancer-associated fibroblasts (CAFs):** CAFs encompass fibroblasts and myofibroblasts comprised within the tumoral stroma. Given the lack of distinctive biomarkers, their location, morphology, and absence of epithelial, endothelial, or leukocyte markers are used to identify such cells within the tumor microenvironment (TME) [81,82].

The biology of fibroblasts comprises a diverse array of functional states, ranging from quiescent fibroblasts, which may be activated by physical stimuli, as in the case of wound healing, to activated myofibroblasts. The latter are characterized by distinct secretome and peculiar markers, such as fibroblast-specific protein 1 (FSP-1), α-smooth muscle actin (α-SMA), desmin, and fibroblast activation protein (FAP).

In this setting, tumors impinge on CAFs distinct signals related to a non-resolutive wound healing process [83]. Molecules such as TGFβ and PDGF are abruptly expressed by both cancer and non-cancer cells to recruit CAFs and activate them to a comparable state as myofibroblasts [84]. In the TME, the most abundant source of CAF is hence thought to be represented by the nearby stroma, even though other sources have been recognized, such as bone marrow-derived mesenchymal stem cells, adipose-derived precursors, pericytes, tissue-resident fibroblasts, and epithelial/endothelial cells undergoing epithelial- to- mesenchymal transition (EMT) or endothelial- to- mesenchymal transition (EndMT) [82,85,86,87,88,89].

Importantly, CAFs are considered key cells in assisting tumor growth. The first evidence supporting their role in tumorigenesis was derived from murine models, in which the addition of fibroblasts to human-induced mammary malignant epithelial cells was able to reduce one-third of the time for tumor development [90]. Moreover, questions arise as to whether CAFs possess distinct properties compared to non-tumoral fibroblasts: mixed tumor graft containing CAFs with human prostate cancer cells determined the growth of tumor masses 500-times larger than tumor grafts containing normal fibroblasts [91]. Indeed, distinctive secretomes between tumoral and non-tumoral fibroblasts have been observed [92]. Moreover, CAFs are able to co-evolve with their neoplastic neighbors by altering their genomes to adapt to the physiologic stresses present within tumors, as observed by dissimilar genomic alterations encountered in CAFs and malignant cells across several tumors [93].

Single-cell studies in preclinical models and human samples led to the definition of three main CAF sub-categories: myofibroblastic (with ECM remodeling and contractile functions), inflammatory/immune-regulatory (with activating and/or suppressive functions), and antigen-presenting CAFs (with immune-engaging properties) (Table 2) [89]. In these regards, CAFs’ adjuvant role in tumor development can be reconducted into a broad spectrum of functions. First of all, a key step in tumor development is represented by the invasion of epithelial cells beyond the basal membrane (BM). In this regard, CAFs are thought to ease this process by the release of matrix-metalloproteinase (MMP) [82] and mechanical forces via integrin-mediated adhesions [94]. CAFs are further able to aid tumor invasiveness by promoting EMT of tumors cells through paracrine signaling [95], as TGFβ–SMAD [96], stroma-derived factor-1 (SDF1), and hepatocyte growth factor (HGF) [97].

As with other cells composing the TME, CAFs undertake bidirectional crosstalk with cancer and non-tumoral cells throughout the production of cytokines and metabolites. In murine models, inactivation of inhibitory TGFβ type II receptors on fibroblast induced a hyperproliferative state of the normal epithelium, ultimately leading to carcinoma development [98]. Different cytokines are secreted by CAFs to promote cancer growth, such as TGFβ, leukemia inhibitory factor (LIF), and HGF [82]. Moreover, cancer growth is also assisted by the buildup of complex metabolic interactions between cancer cells and CAFs [99]: for example, in pancreatic cancer autophagy of CAFs has been observed to provide tricarboxylic acid cycle metabolites to assist tumor cells growth [100].

Furthermore, CAFs can mediate therapy resistance via drug scavenging mechanisms, the production of exosomes [101], or the release of cytokines, as for HGF-driven collateral ERK-MAPK pathways activation upon BRAF inhibition [102]. In addition, CAFs have also been shown to contrast ROS production via glutathione upregulation [103], and to support cancer cell stemness via the production of IL-6 and IL-8 [104].

CAFs also mediate angiogenesis through different mechanisms: they are able to produce high concentrations of SDF-1/CXCL12, which are able to recruit endothelial precursor cells (EPC) into TME; their subsequent production of VEGF induces subsequently the latter to differentiate to form neovasculature [105].

CAFs also possess a central role in immune regulation [106] via the production of immunosuppressive cytokines, such as CXC-ligand 9 (CXCL9) and TGFβ, by operating as antigen-presenting cells, and by co-expressing PD-L2 and FASL, ultimately dampening adaptive immunity [107]. Indeed, an inverse correlation exists between CAFs and CD8+ T-cells concentration in the TME [108]. In pancreatic cancer, leucine-rich repeat-containing 15 (LRRC15) positive CAFs, induced by TGFβ, have been associated with inferior survival upon ICB [109]. In these regards, LRRC15 targeting by Samrotamab vedotin, ABBV-085, an anti-LRRC15 monomethyl auristatin E (MMAE)-loaded antibody-drug conjugate (ADC), has shown robust preclinical activity and has been tested in phase I clinical trial in patients with advanced solid tumors, showing tolerability and preliminary activity [110,111]. Another molecular pathway identified in myofibroblastic, as opposed to inflammatory, CAFs in pancreatic cancer is sonic hedgehog (SHH) signaling. Interestingly, its inhibition by the SHH antagonist LDE225 (sonidegib) has been shown to delay tumor growth and alter the TME composition by increasing inflammatory CAFs as well as CD8 lymphocytes [112]. Conversely, inflammatory CAFs are induced by IL1-JAK-STAT-related signaling, which is induced by TNFa and IL1 and counterbalanced by TGFβ [113].

Despite being considered key pro-tumorigenic cells, tumor-suppressive functions of CAFs have also been observed. As an example, high concentrations of CD146+ CAFs confer prolonged tamoxifen responsiveness in breast cancer [114]. Furthermore, in PDAC murine models, the elimination of SMA+ CAFs demonstrated to lead to disease exacerbations [115,116]. This further sustains the existence of different subtypes of CAFs, with important therapeutical repercussions in the view of CAFs targeting treatment approaches.

**Extracellular matrix (ECM):** The ECM represents the biological milieu hosting cells, and it is physiologically composed of more than 300 molecules including collagens, proteoglycans, and glycoproteins. The ECM is considered to be a dynamic three-dimensional network [117] affecting cellular growth, survival, motility, and differentiation [118]. ECM modifications occur during cancer development and progression [119], with cancer and stromal cells able to provide EMT modifications in different ways.

A characteristic of the tumor stroma is represented by its abundant molecular architecture, conferring a peripheral structural stiffness known as a desmoplastic reaction. ECM deposition is mainly performed by myofibroblast, as with physiological wound repair. In the TME, instead, sustained cytokines released by cancer and non-cancer cells, such as TGFβ, abruptly activate CAFs leading to ECM deposition [117]. Tissue stiffening is also supported by crosslinking enzymes, produced mainly by stromal cells [120]. The desmoplastic stroma has long been associated with poor prognosis [121], and the inhibition of stromal stiffening by inhibiting cross-linking enzymes was shown to decrease tumor metastasis in breast cancer pre-clinical models [122]. Indeed, the desmoplastic reaction is able to favor tumor growth in different ways: in the first instance, a denser tumor stroma enhances the so-called mechanosignaling, whereupon direct interactions occur between cancer and stromal molecules lead to increased integrin-regulated pro-survival signaling [123], and cancer cell motility [124]. ECM molecules also directly affect tyrosine kinase receptors: Discoidin Domain Receptors (DDR1 and DDR2) are activated upon collagen binding, leading to enhanced tumor growth and stemness signaling via STAT3 activation [125]. Tyrosine kinase inhibitors dasatinib and nilotinib, due to their ability to target DDR1 and 2, have been tested in patients with advanced solid tumors harboring activating mutations in DDR1 or 2 (NCT01514864 and NCT02029001, respectively), although yielding poor efficacy results [126]. Moreover, adhesion-dependent Fak phosphorylation stimulates Ras and PI3K signaling, ultimately enhancing MAPK downstream signaling [127]. In addition, the desmoplastic reaction negatively influences drug sensitivity: the stiffened matrix determines a physical barrier to drug delivery via physical impediment and microvessels compression [128]. Moreover, matrix stiffness is also linked to radiotherapy resistance [129]: indeed, higher β1 integrin expression is upregulated after radiotherapy exposure, and it showed to confer radiation resistance to cancer cells [130], as inhibition of integrin signaling has been shown to increase sensibility to radiation treatments in human xenograft models [131].

Other than quantitative adaptations, qualitative changes also occur. Indeed, it has been observed that a linearized and perpendicular disposition of collagen fibers confers a higher invasiveness capacity in breast cancer cells [132]. In a preclinical model of pancreatic cancer, it has been shown that cancer-secreted type I collagen homotrimers interact with a3b1 integrin on cancer cells promoting cell pro-survival and proliferative signaling cascades, and their inhibition leads to increased tumoral T cell infiltration and enhanced anti-cancer responses upon anti-PD1 immunotherapy [133]. Integrin targeting, either directly or by interfering with downstream effectors (i.e., Src and FAK) is currently also being investigated in combination with ICB [126].

Another modification occurring in tumor ECM is the proteolytic degradation elicited by enzymes secreted mainly by stromal cells. In such a manner, migratory tracks are created aiding cancer cells’ movement by decreasing mechanical stress [134]. Moreover, ECM physiologically contracts interactions with different soluble molecules, such as growth factors, hence providing, once released by proteolytic degradation, growth stimuli for cancer cells. For example, fibronectin binds with high affinity to molecules such as insulin-like growth factor binding-protein-3, FGF-2, and VEGF-A [135]. Another glycoprotein that acts as a binding site for other chemokines/cytokines is represented by CD44, in particular, the variant (CD44v) and the hyaluronic acid-induced (HA-CD44) isoforms. Numerous trials are investigating the safety and efficacy of anti-CD44 mAb (NCT01358903), as well as vCD44-specific mAb, or in the form of antibody drug conjugates (Bivatuzumab-mertansine) (NCT02254031, NCT02254005, NCT02254018, NCT02254044) [126].

Interestingly, the cleavage of long ECM molecules leads to the release of shorter fragments with structures reminiscent of cytokines, hence defined as matrikines [136]. Matrikines have been observed to have both pro-tumorigenic and anti-tumor functions: for instance, NC1 domains of basal membranes’ collagen exert an antiangiogenic function [137]; the α3-chain of collagen IV, by inhibiting MMP, hampered tumor cells invasiveness. On the other hand, elastin-derived matrikines have been demonstrated to promote tumor progression [138].

ECM remodeling represents a critical step in the development of tumor metastasis. Dissimilar changes occur in primary compared to secondary neoplastic sites: while a prominent collagen deposition prevails in primary tumors, metastatic sites are mainly composed of fibronectin, with high deposition observed also in the pre-metastatic niche to aid tumor seeding [139]. As an example, fibronectin secreted by endothelial cells was revealed to assist circulating tumor cells (CTC) in extravasating to the new homing stroma [140]. Furthermore, ECM remodeling was demonstrated to be able to reactivate dormant metastatic cells: in mouse models, the production of proteolytically remodeled laminin was able to reactivate the latter by activating integrin α3β1 signaling [141]. ECM molecules are also expressed by CTC, which are potentially able to sustain tumor survival signaling in adverse conditions and to protect their clearance from blood-filtering immune barriers [142].

In addition, ECM alterations also occur in response to anti-cancer therapies. For instance, radiotherapy (RT) has been shown to induce TGFβ expression within the TME, leading to CAF activation, EMT, and PD-L1 upregulation in cancer cells. This process has been modulated in pre-clinical models with a bispecific functional protein, bintrafusp alpha (BA), which acts both as a TGFβ trap as well as an anti-PDL1 agent [143]. Indeed, in murine models, the combination of BA with RT led to decreased tumor infiltrate of myeloid-derived suppressor cells (MDSCs), CAF activation and ECM deposition, to increased CD8/Treg ratio, and overall survival of control mice or upon monotherapies. In line with these results, BA is currently being tested in several clinical trials. HCW9218 instead is a bifunctional TGFβ trap coupled to IL15 protein, which has shown potent preclinical activity by augmenting immune responses and it is currently being tested in patients with advanced solid (NCT05322408) or pancreatic tumors (NCT05304936) [144]. Other drugs targeting TGFβ comprise either TGFβ-traps (i.e., AVID200 or luspatercept), small tyrosine kinase inhibitors (i.e., fresolimumab), antisense oligonucleotides (i.e., AP12009) as well as anti-TGFβ mAb (i.e., vactosertib or galunisertib) [145].

Overall, ECM is subjected to different changes involving its topographical disposition, chemical properties, and direct interactions with soluble factors or membrane-bound receptors, ultimately contributing in different ways to the promotion of tumor development, growth, and resistance to antitumor therapies.

## 4. Discussion

Solid tumors are constituted by several different cellular and extracellular components, known as the Tumor Microenvironment (TME). The relationship established between cancer cells and the healthy surrounding TME often dictates the natural fate of cancer lesions, from tumor eradication to overt tumor progression, immune evasion, and/or therapy resistance. Malignant tumors typically hijack the healthy, non-tumoral, TME, which in turn has been recognized as a key therapeutic target at multiple levels. Immune cells, such as T lymphocytes and myeloid cells, as well as stromal cells and extracellular matrix components, are crucial recognized components of the TME. By dissecting the functional states and molecular players at the base of TME derailment, numerous therapeutic targets have been elucidated and some have reached clinical approval in several solid tumors, such as immune checkpoint blockers (ICB) (Figure 2). Moreover, ICB, in the form of anti-PD1/anti-PDL1, has also been approved in the early, neo-adjuvant, setting in TNBC [146] as well as NSCLC [147]. Clinical research is nowadays moving towards the identification of the ideal clinical settings as well as combinatorial partners to combine ICB with, in order to promote durable anti-cancer responses. Notwithstanding this, tumor relapse or therapy refractoriness remain a major issue in oncology upon TME-targeting agents, especially upon myeloid-targeting. In particular, a major hurdle limiting the clinical efficacy of macrophage-depleting strategies is represented by their remarkable functional heterogeneity, which in turn dictates differential roles of these cells according to specific organs/tumors, or within a single tumor lesion. In addition to this, species-specific differences have been demonstrated among murine and human macrophages, further explaining the meager clinical trial results obtained so far despite promising pre-clinical attempts [148]. Concurrently, on the one hand, myeloid-depleting strategies have so far found a role in TGCT with paxidartinib, on the other hand, myeloid-reprogramming strategies are currently also being explored to promote long-lasting and effective anti-tumor immune responses (NCT03866109) [149,150], or to provide novel effector functions, such as for chimeric antigen receptors-macrophages (NCT04660929) [151]. Ultimately, CAF targeting is still a largely unexplored area of clinical investigation, due to its great heterogeneity, less characterized functional adaptations, and the scarcity of specific, druggable, targets.

## 5. Conclusions

The medical treatment of solid tumors has vastly changed in the past decades from the sole use of chemotherapeutic agents to the introduction of both more specifically targeted and immune therapies. Concerning immune therapies, the most pronounced clinical benefits have been observed with immune checkpoint blockers (anti-PD1/PDL1, anti-CTLA4 mAb), despite multiple cellular players comprising the TME and aiding in tumor progression and therapy resistance. In this scenario, the identification of novel druggable TME vulnerabilities, mainly focusing on T lymphocytes, macrophages, and CAFs, have been elucidated and are currently in clinical testing. Information deriving from ongoing clinical trials will possibly expand the therapeutic armamentarium targeting the TME, apart from ICBs, and/or provide novel combinatorial strategies. In addition, novel technologies able to study cancer lesions and their TME at single-cell resolution are nowadays available, allowing to match DNA-, RNA- and/or protein-based approaches. In addition, in vitro 3D modeling of TME has been shown to mimic in vivo drug delivery, deepen the diverse functionalities of CAFs [152,153] and macrophages [154], and represent a valuable tool for drug screening [155,156]. Overall, the identification of novel molecular drivers of cancer progression and TME derangement will set the bases for newer therapeutic targets. 

## Figures and Tables

**Figure 1 cancers-14-04278-f001:**
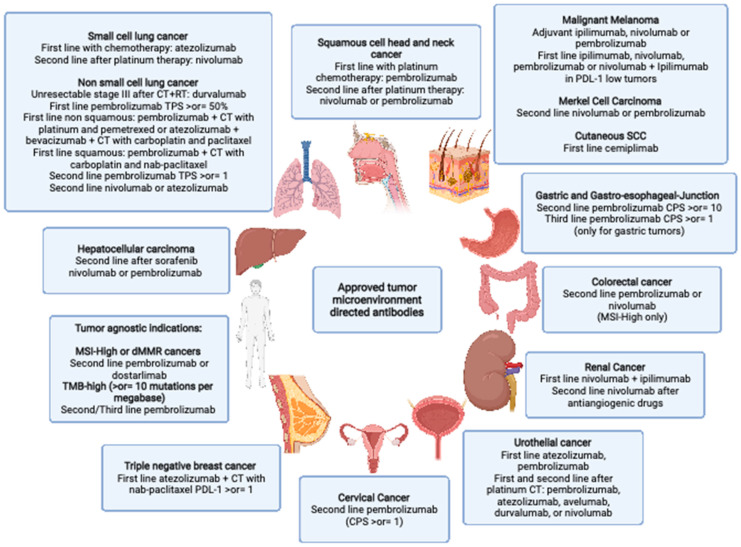
Approved tumor microenvironment directed antibodies: SCC: squamous cell carcinoma SCHNC: squamous Cell Head and Neck Cancer SCC: small cell lung cancer NSCLC: non small cell lung cancer CT: chemotherapy RT: radiotherapy TNBC: triple-negative breast cancer GEJ: gastroesophageal junction dMMR: mismatch repair deficient) MSI-H: microsatellite instability-High CRC: colorectal cancer TMB: tumor mutational burden. Created with BioRender.com.

**Figure 2 cancers-14-04278-f002:**
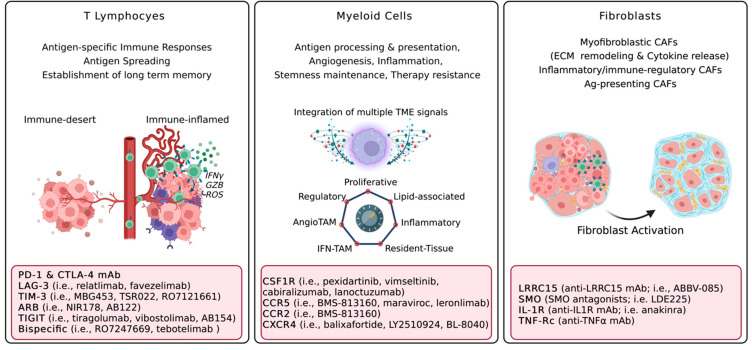
Major pro-tumorigenic functions of cellular TME components and therapeutic vulnerabilities. Acronyms: Interferon gamma, IFNγ; granzyme, GZB; Reactive Oxygen Species, ROS; Programmed Death 1, PD-1; Lymphocyte Activation Gene 3, LAG-3; T-cell immunoglobulin and mucin domain 3, TIM-3; Adenosine Receptor Blockers, ARB; T cell immunoreceptor with Ig and ITIM domains, TIGIT; Tumor Microenvironment, TME; Tumor-Associated Macrophages, TAM; colony-stimulating factor 1 receptor, CSF1R; C-C chemokine receptor, CCR; C-X-C chemokine receptor type 4, CXCR4; Extracellular Matrix, ECM; Cancer-associated fibroblasts, CAFs; LRRC15, Leucine Rich Repeat Containing 15; Smoothened, SMO; Interleukin 1 Receptor, IL-1R; monoclonal antibody, mAb; Tumor necrosis factor alpha, TNFα. Created with BioRender.com.

**Table 1 cancers-14-04278-t001:** Selected clinical trials with new immune checkpoint inhibitors.

Drug	Target	Disease	Study Phase	Study Drugs	NCT Trial Identifier
**Relatlimab**	LAG-3	Melanoma	Phase I	Relatlimab + Ipilimumab in pts progressing on anti-PD1	NCT03978611
		Melanoma	Phase II	Nivolumab vs. Relatlimab vs Nivolumab + Relatlimab in immunotherapy naïve pts	NCT03743766
		Uveal Melanoma	Phase II	Nivolumab + Relatlimab in untreated pts	NCT04552223
		NSCLC	Phase II	Relatlimab + Nivolumab + CT vs. Nivolumab + CT in untreated pts	NCT04623775
		HNSCC	Phase II	Nivolumab vs. Nivolumab + Relatlimab or Nivolumab + Ipilimumab	NCT04080804
		CRC		Nivolumab + Relatlimab in pretreated pts	NCT03642067
		Hepatocellular carcinoma	Phase II	Relatlimab + Nivolumab in pts progressing to TKIs	NCT04567615
**Favezelimab**	LAG-3	CRC	Phase III	Favezelimab + Pembrolizumab vs SOC in PD-L1 + pretreated pts	NCT05064059
**Tebotelimab**	LAG-3 + PD1	HNSCC	Phase II:	Enoblituzumab + Retifanlimab vs. Tebotelimab	NCT04634825
**Tiragolumab**	TIGIT	SCLC	Phase III SKYSCRAPER-02	Atezolizumab + CT + Tiragolumab vs. Atezolizumab + CT in untreated pts	NCT04256421
	TIGIT	NSCLC	Phase III SKYSCRAPER-01	Tiragolumab + atezolizumab vs. atezolizumab in PDL1 high untreated pts	NCT04294810
	TIGIT	NSCLC	Phase III SKYSCRAPER-03	Tiragolumab + Atezolizumab vs. durvalumab in adjuvant setting stage III	NCT04513925
	TIGIT	NSCLC	Phase II SKYSCRAPER-05	Tiragolumab + Atezolizumab CT vs. Tiragolumab + Atezolizumab locally advanced untreated pts	NCT04832854
**LY3415244** ^§^	TIM-3+PD-L1	Solid tumors	Phase I	LY3415244 in pretreated pts	NCT03752177
**TSR022**	TIM-3	Solid tumors	Phase I	TSR022 vs. TSR022 combined with CT or other checkpoint inhibitors in pretreated pts	NCT02817633
**NIR178**	A2AAR inhibitors	Solid tumors	Phase I	NIR178 vs. NIR178 + antiPD1 PDR001	NCT02403193
**AB-928**	A2AAR inhibitors	Solid tumors	Phase I/II	AB-928 + antiPD1 ziberelimab + FOLFOX ± Bevacizumab vs. CT ± Bevacizumab vs. regorafenib in CRC pretreated pts	NCT04660812

^§^ Bispecific antibody binding two distinct immunomodulating targets. Acronyms: Non small cell lung cancer, NSCLC; small cell lung cancer, SCLC patients, pts; colorectal cancer, CRC; Head and neck squamous cell carcinoma, HNSCC; Programmed Death 1, PD-1; Programmed Death 1 Ligand, PD-1L; Lymphocyte Activation Gene 3, LAG-3; T-cell immunoglobulin and mucin domain 3, TIM-3; T cell immunoreceptor with Ig and ITIM domains, TIGIT; A2A Adenosine Receptor, A2AAR; chemotherapy, CT; standard of care, SOC.

**Table 2 cancers-14-04278-t002:** Major cancer-associated fibroblast subtypes within the tumor microenvironment. The table summarizes the chief characteristics and putative markers of major CAF subtypes. Acronyms: extracellular matrix, ECM; collagen, COL; interleukin, IL; transforming growth factor, TGF, Leucine Rich Repeat Containing 15, LRRC15; human leukocyte antigen II, HLA-II; Leukemia inhibitory factor, LIF; Granulocyte-Macrophage Colony-Stimulating Factor, GM-CSF; C-C Motif Chemokine Ligand, CCL; C-X-C Motif Chemokine Ligand, CXCL; complement C1q chain, C1; cluster of differentiation, CD.

	Myofibroblastic	Inflammatory/Immune-Regulatory	Antigen-Presenting
**Characteristics**	Collagen production ECM remodeling (i.e., COL1A1, COL10A1, COL4A1, MMP3) Tumor-promoting cytokine secretion (i.e., IL4, IL13, TGF-β) Cancer cell growth & metastasis	Immune cell recruitment/regulation (i.e., IL-6, IL-11, IL-8, LIF, GM-CSF, CXCL1, CXCL2, CXCL8, CXCL10, CXCL12, CCL2, CCL8) Inflammation and complement regulation (i.e., C1QA, C1QB, C1QC) Cancer cell activation	T lymphocyte engagement, recruitment, activation (i.e., HLA-II, CXCL12, CXCL14)
**Putative markers**	COL4A1, LRRC15	CD63	HLA-II, CD105lo

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
