# Peer review of "Targeting Cellular Components of the Tumor Microenvironment in Solid Malignancies"

_cancers, 2022, doi:10.3390/cancers14174278_

Round 1
Reviewer 1 Report
In this review article, Belli et al. discuss the main components of the tumor microenvironment and provide a broad overview of therapeutic approaches targeting these components in various solid malignancies. The manuscript is well-structured and provides an accurate and current representation of the topic. However, some improvements should be made in order to improve the manuscript.
Major comments:
● Because the scope of this review article is very broad, the article lacks a clear framework for examining the topic. For example, section 1 of the article discusses therapeutic approaches targeting T cell activation, whereas section 2 deals with myeloid cell recruitment but not activation. Furthermore, section 3 provides only limited discussion of recent progress in therapeutic targeting of CAFs and instead chooses to focus on basic CAF biology and ECM organization. Because the topic chosen by the authors is too broad to be manageable, the authors should narrow down the focus of their review. For example, the authors may choose to focus on modulating immune cell recruitment into the TME.
● Since the article mainly focuses on targeting, and not profiling the TME, the title “Cellular Profiling of the tumor Microenvironment in Solid Malignancies” is somewhat inaccurate. A title that is more in line with the article’s main topic (i.e. therapeutic targeting of the TME) would be a better choice.
● The discussion section is too short and should include a more extensive and critical discussion of the information presented in the main body. For example, the authors could choose to focus on the remaining gaps in our understanding of myeloid cell recruitment into the TME and why therapies such as anti-CSF1 mAbs demonstrated only limited survival benefit in clinical trials.
● CXCR4 inhibitors are widely published for their vascular- and ECM-modulating effects; these are not discussed here.
Minor comments:
● The claim at line 19 in the abstract that most immune cells are recruited from circulation is only true in some cancers (in many tumors, resident immune cells dominate); clarification should be provided.
● Since in addition to focusing on therapeutic approaches targeting the TME the article also provides general discussion of its cellular components, the sentence in line 105 is somewhat misleading.
● Although figures are easy to understand and complement the content of the review, they should be referred to in the main text of the article.
● It would be helpful for the reader if a brief explanation of margetiximab was provided in line 167.
● The manuscript is written in proper academic English and is free of significant errors that would impede understanding. However, some minor errors include:
○ The sentence in lines 45-48 is convoluted.
○ In line 221, it is unclear whether 85 patients who experienced partial response received a combination treatment.
○ In line 233, “activation” should be replaced with “activating”.
○ ECM is sometimes mistyped as EMC, e.g., line 296
● This review would benefit from a graphical abstract.
Author Response
Major comments:
- Because the scope of this review article is very broad, the article lacks a clear framework for examining the topic. For example, section 1 of the article discusses therapeutic approaches targeting T cell activation, whereas section 2 deals with myeloid cell recruitment but not activation. Furthermore, section 3 provides only limited discussion of recent progress in therapeutic targeting of CAFs and instead chooses to focus on basic CAF biology and ECM organization. Because the topic chosen by the authors is too broad to be manageable, the authors should narrow down the focus of their review. For example, the authors may choose to focus on modulating immune cell recruitment into the TME.
We thank the reviewer for the comment, we have edited the manuscript abstract, introduction and to state more clearly the focus of the review, and we have expanded the ECM-targeting section as well.
- Since the article mainly focuses on targeting, and not profiling the TME, the title “Cellular Profiling of the tumor Microenvironment in Solid Malignancies” is somewhat inaccurate. A title that is more in line with the article’s main topic (i.e. therapeutic targeting of the TME) would be a better choice.
We thank the reviewer for the comment, we have changed the manuscript title accordingly:
OLD: Cellular Profiling of the Tumor Microenvironment in Solid Malignancies
NEW: Targeting Cellular Components of the Tumor Microenvironment in Solid Malignancies
- The discussion section is too short and should include a more extensive and critical discussion of the information presented in the main body. For example, the authors could choose to focus on the remaining gaps in our understanding of myeloid cell recruitment into the TME and why therapies such as anti-CSF1 mAbs demonstrated only limited survival benefit in clinical trials.
We thank the reviewer for the comment, we have added to the text sections and references regarding mechanisms at the base of anti-CSF1 mAb failures. Moreover, we have broadened the discussion and added a conclusion part, as requested.
- CXCR4 inhibitors are widely published for their vascular- and ECM-modulating effects; these are not discussed here.
We thank the reviewer for the comment, we have edited the manuscript as follows:
OLD: The CXCR4-CXCL12 axis is involved in multiple cancer processes, including signaling pathways related to cellular proliferation (PI3K/Akt and Ras/Raf/MAPK) (63) (64), immune-suppression (65) (66) (67), metastatic spreading (68) (69).
NEW: The CXCR4-CXCL12 axis is involved in multiple cancer processes, including signaling pathways related to cellular proliferation (PI3K/Akt and Ras/Raf/MAPK) (63) (64), immune-suppression (65) (66) (67), metastatic spreading (68) (69) and angiogenesis (70). Concerning the latter, it has been shown that CXCR4-CXCL12 axis inhibition curbs vasculogenesis in pre-clinical models through AKT-mediated signalling pathway leading to VEGF induction. In addition, in mice, CXCR4 acts as a chemoattractant for circulating neutrophils to the liver, aiding in the formation of pre-metastatic niches (71).
Minor comments:
- The claim at line 19 in the abstract that most immune cells are recruited from circulation is only true in some cancers (in many tumors, resident immune cells dominate); clarification should be provided.
We thank the reviewer for the comment, we have edited the manuscript accordingly.
- Since in addition to focusing on therapeutic approaches targeting the TME the article also provides general discussion of its cellular components, the sentence in line 105 is somewhat misleading.
We thank the reviewer for the comment, we have deleted the misleading sentence.
- Although figures are easy to understand and complement the content of the review, they should be referred to in the main text of the article.
We thank the reviewer for the comment, we have inserted figures and tables references in the text.
- It would be helpful for the reader if a brief explanation of margetiximab was provided in line 167.
We thank the reviewer for the comment, we have edited the manuscript as follows:
OLD: Tebotelimab (MGD013): belongs to bispecific moAb targeting both LAG-3 and PD-1 and it was tested in a phase I clinical trial alone or in combination with margetuximab in case of HER2 expression in advanced or metastatic solid or hematologic malignancies (15).
NEW: Tebotelimab (MGD013): belongs to bispecific moAb targeting both LAG-3 and PD-1 and it was tested in a phase I clinical trial alone or in combination with margetuximab (an anti-HER2 mAb with higher binding activity for activating Fcγ receptor CD16A and decreased affinity for inhibitory CD32B) in case of HER2 expression in advanced or metastatic solid or hematologic malignancies (15).
- The manuscript is written in proper academic English and is free of significant errors that would impede understanding. However, some minor errors include:
We thank the reviewer for the comment, we have thoroughly revised English throughout the text.
○ The sentence in lines 45-48 is convoluted.
We thank the reviewer for the comment, we have modified the sentence as follows:
OLD: Tumor cells metastasizing in distant tissues can survive and expand in normal mi- 45 croenvironment or meet a favorable microenvironment (pre-metastatic niches), positively 46 conditioned before the arrival of tumoral cells from several conditions such as circulating 47 factors released by primary tumor (1) (2).
NEW: Tumor cells metastasizing to distant tissues may meet a favorable microenvironment (pre-metastatic niche), which is positively conditioned by circulating factors released by the primary tumor (1) (2).
○ In line 221, it is unclear whether 85 patients who experienced partial response received a combination treatment.
We thank the reviewer for the comment, we have modified the sentence.
○ In line 233, “activation” should be replaced with “activating”.
We thank the reviewer for the comment, we have modified the sentence.
○ ECM is sometimes mistyped as EMC, e.g., line 296
We thank the reviewer for the comment, we have modified the typo.
- This review would benefit from a graphical abstract.
We thank the reviewer for the comment, we have added the graphical abstract and have provided for a review summary as GA caption.
Reviewer 2 Report
1.
This review is informative and valuable for researchers on cancers. The authors should add a section on 3D cancer-TME modeling in vitro to investigate the new drug candidates. The section added would lead to a better review.
Review (for concept)
Cancers 12 (10), 2754
http://doi.org/10.1089/ten.teb.2009.0676
Research
For CAF
doi.org/10.1016/j.actbio.2018.05.055
Tissue Eng. Part C Methods 2019, 25, 711-720. https://doi.org/10.1089/ten.tec.2019.0189
https://doi.org/10.1002/adhm.201600772
For TAM
Cells 2022, 11(9), 1583
2.
The authors should add the table for characteristics of several stromal cells.
Author Response
- This review is informative and valuable for researchers on cancers. The authors should add a section on 3D cancer-TME modeling in vitro to investigate the new drug candidates. The section added would lead to a better review. Review (for concept): Cancers 12 (10), 2754; http://doi.org/10.1089/ten.teb.2009.0676 Research: for CAF doi.org/10.1016/j.actbio.2018.05.055 , Tissue Eng. Part C Methods 2019, 25, 711-720. https://doi.org/10.1089/ten.tec.2019.0189 , https://doi.org/10.1002/adhm.201600772; for TAM Cells 2022, 11(9), 1583
We thank the reviewer for the comment, we have added this section and references to main text (conclusion) as follows:
In addition, also in vitro 3D modelling of TME has been shown to mimic in vivo drug delivery, deepen the diverse functionalities of CAFs (147) (148) and macrophages (154), and represents a valuable tool for drug screening (155) (156).
- The authors should add the table for characteristics of several stromal cells.
We thank the reviewer for the comment, we have added table 2 to main text.
Reviewer 3 Report
The authors of this review article seek to present a comprehensive review of the tumor microenvironment factors contributing to tumor progression. Overall, they do a nice job of presenting a multifaceted review of multiple components contributing to tumor progression with a particular focus on T cells and myeloid (mostly macrophages) in terms of therapeutic development. The work would certainly be acceptable and a contribution to the field with some minor revisions.
In particular, the abstract states it seeks to be a coprehensive review, but really focuses on T cells and T cell therapies and the same goes for macrophages in the myeloid compartment. It occasionally addresses other cells such as NKs and DCs, but for the most part, these are brief references regarding function rather than therapeutics which have been developed targeting these immune cell subsets. I believe by setting the focus from the start (abstract) and continuing this focus rather than a broad coverage of all the immune system, the authors could still make a very significant contribution while meeting their stated aims presented in the abstract. The authors do a very nice job of summarizing clinical trial data available for investigational therapies as well.
Comments:
1. I would narrow the aims stated in the introduction (Line 52) to fit the strong points of data presented in this manuscript. T-cells are clearly covered in depth as are some myeloid cells. NK cells are conspicuously mentioned, but then no further information regarding therapies focused on NK targeting of the TME is expanded upon.
2. In general, grammar and syntax is good. There are multiple missing articles (a, the) throughout the manuscript, which do not change meaning, but are distracting to the reader. Please consider additional edits to address.
3. Numerous plural/possessive errors which again do not change the meaning of sentences, but are distracting to the reader. Please edit.
4. Table 1, I would consider leaving out trials that are actively discussed in the body of the manuscript. This is covered and doesn't need to be repeated in the table. Consider including only trials without outcome data in the table as this is not covered in the body of the manuscript, but gives additional data to highlight the pending work in the area.
5. TGB-beta is mentioned, as is one brief reference of TGF-beta traps or blockade. This is really a growing area of TME manipulation that has multiple options under investigation and impacts lymphocytes and granulocytes. This should be addressed and expanded upon.
6. The tone and content of the manuscript changes considerably in the sections discussing the ECM. The focus really goes more to mechanism and drifts away from therapeutic interventions focused on in earlier parts of the manuscript. There should be an address of ECM targeting therapies as well to keep with the overall theme an focus of the authors in earlier parts of the review.
7. The discussion is rather brief given the interest and growth in the field of TME manipulation.
8. Multiple drugs are listed, but are missing mechanism (ex linen 369 and 376, but not all-inclusive). Please clarify whether these are agonists, antagonists or work as a "sink" for the cytokine or chemokine. This would be helpful for the reader.
Author Response
- I would narrow the aims stated in the introduction (Line 52) to fit the strong points of data presented in this manuscript. T-cells are clearly covered in depth as are some myeloid cells. NK cells are conspicuously mentioned, but then no further information regarding therapies focused on NK targeting of the TME is expanded upon.
We thank the reviewer for the comment, we have edited the manuscript abstract, introduction and to state more clearly the focus of the review, with a particular focus on T cells, macrophages, and cancer associated fibroblasts.
- In general, grammar and syntax is good. There are multiple missing articles (a, the) throughout the manuscript, which do not change meaning, but are distracting to the reader. Please consider additional edits to address.
We thank the reviewer for the comment, we have thoroughly revised English throughout the text.
- Numerous plural/possessive errors which again do not change the meaning of sentences, but are distracting to the reader. Please edit.
We thank the reviewer for the comment, we have thoroughly revised English throughout the text.
- Table 1, I would consider leaving out trials that are actively discussed in the body of the manuscript. This is covered and doesn't need to be repeated in the table. Consider including only trials without outcome data in the table as this is not covered in the body of the manuscript but gives additional data to highlight the pending work in the area.
We thank the reviewer for the comment, we have amended Table 1 according to the reviewer’s comment.
- TGB-beta is mentioned, as is one brief reference of TGF-beta traps or blockade. This is really a growing area of TME manipulation that has multiple options under investigation and impacts lymphocytes and granulocytes. This should be addressed and expanded upon.
We thank the reviewer for the comment, we have added a new section regarding TGF-b targeting part as requested:
“HCW9218 instead is a bifunctional TGFβ trap coupled to IL15 protein, which has shown potent preclinical activity by augmenting immune responses and it is currently being tested in patients with advanced solid (NCT05322408) or pancreatic tumors (NCT05304936) (144). Other drugs targeting TGFβ comprise either TGFβ-traps (i.e., AVID200 or luspatercept), small tyrosine kinase inhibitors (i.e., fresolimumab), antisense oligonucleotides (I.e., AP12009) as well as anti-TGFβ mAb (i.e., vactosertib or galunisertib) (145).”
- The tone and content of the manuscript changes considerably in the sections discussing the ECM. The focus really goes more to mechanism and drifts away from therapeutic interventions focused on in earlier parts of the manuscript. There should be an address of ECM targeting therapies as well to keep with the overall theme and focus of the authors in earlier parts of the review.
We thank the reviewer for the comment, we have added new sections regarding ECM targeting in the manuscript.
- The discussion is rather brief given the interest and growth in the field of TME manipulation.
We thank the reviewer for the comment, we have expanded the discussion section and added a conclusion paragraph.
- Multiple drugs are listed, but are missing mechanism (ex linen 369 and 376, but not all-inclusive). Please clarify whether these are agonists, antagonists or work as a "sink" for the cytokine or chemokine. This would be helpful for the reader.
We thank the reviewer for the comment, we have added clarifications for missing mechanisms of action.
Round 2
Reviewer 1 Report
The authors have addressed the previous concerns of this reviewer.
Reviewer 2 Report
OK.